# Ion Transport in Solid Medium—Evaluation of Ionic Mobility for Design of Ion Transport Pathways in Separator and Gel Electrolyte

**DOI:** 10.3390/membranes11040277

**Published:** 2021-04-09

**Authors:** Yuria Saito

**Affiliations:** 1Separator Design Co. Ltd. 1725-1, Hagyu, Ooaza, Iide-machi, Yamagata 999-0602, Japan; yuria-saitou@separator-design.com; Tel.: +81-238-88-7755; 2National Institute of Advanced Industrial Science & Technology 1-8-31, Midorigaoka, Ikeda 563-8577, Japan

**Keywords:** separator membrane, polymer gel electrolyte, ionic mobility, microviscosity, Coulombic interaction, restricted diffusion

## Abstract

Further improvement in the performance of lithium secondary batteries will be an indispensable issue to realize a decarbonized society. Among them, the batteries for electric vehicles still have many issues to be addressed because they are subject to various conditions such as high-power performance, safety, and cost restrictions for widespread use. Those subjects require extensive researches from the improvement of each element material to control the battery system to optimize the total performance. Based on this idea, we have been conducting research focusing on ion movement to elucidate the ion conduction mechanism from the microscopic point of view. It has been recognized that the ionic mobility in the battery, which dominates the power performance of the battery, is affected by the solid environment in which the ions move (separator and electrode materials) and the evaluation of ion movement, including the interaction with the surroundings, is necessary as an essential step for battery design. In this article, I will introduce the evaluation approach of ion dynamics and the evaluation results of mobility and interactive situations of carrier ions in the practical separator membranes and gel electrolytes. Finally, the direction of material design is outlined through this review.

## 1. Introduction

Ionic mobility in the battery system determines the power performance of the battery [1,2]. The development of a high-power battery to create a battery in which the high ion mobility is maintained stably during the battery operation is needed in modern society where the depletion of energy resources is concerned.

Most of the ion dynamics evaluations that have been used in battery development so far have been focused on macroscopic properties such as the ionic conductivity of electrolyte materials and the I/V characteristics of batteries. These values are effective in comparing the conductive performances between the different types of materials or batteries. However, they are unlikely to be reasonable indices used in the design and construction of the new elemental material and device of the battery while overcoming the current issues. This is because the macroscopic values do not suggest the cause of mobility that reflects the situation around the carrier ions. In order to develop new high-power batteries rationally, it is necessary to develop the dynamic indices reflecting the factors of mobility and to establish the systematic approach for the estimation of the indices simply and correctly.

This review represents the proposal of new indices of mobility, their estimation methods, and their application to practical materials for evaluations of ion transport mechanism associated with the structures of a separator and gel electrolyte as ion migration fields, that is a summary of the process of establishing evaluation technology leading to material design. We focused on the interactions between the carrier ions and surrounding species and solid media in the battery as dominant factors of ionic mobility. In this review, the separators [3,4,5] and the polymer chains in the gel electrolytes [6,7] were taken up as the typical solid media influencing ion mobility. This is because the separator is the main element of the battery to keep the ions stable and serve the pathways for ion transfer between the electrodes. Further, it is easy to evaluate and control the path structure for ion movement using their structural parameters such as porosity, pore size, and path tortuosity. On the other hand, the structure of polymer gel electrolytes, which were initially developed with the aim of being a self-supporting electrolyte for easy handling, is relatively easy to control based on the polymer type, cross-linked state, and the number of polar sites. Polymer gel electrolytes could replace separators in the near future when it becomes a fully self-supporting structure with high ionic mobility.

To obtain the dynamic values, we used the pulsed gradient spin-echo NMR technique as a key technology in this research for measurements of diffusion coefficients. The technique has the advantage that the diffusion coefficient of each nuclear species (*D*_Li_*, D*_F_*, D*_H_) can be obtained individually. The diffusion coefficients of inherent ionic species, *D*_cation_ and *D*_anion_, were estimated based on the theoretical model originally constructed, using the measured ionic conductivity *σ* and *D*_Li_, *D*_F_, *D*_H_. The interaction, that is, the influence of the surrounding species and media on the mobility of the ions and molecules, is reflected in the microviscosity that the moving ions and molecules feel as resistance. Using the estimated *D*_cation_, *D*_anion_, and *D*_solvent_, different types of microviscosities (η, α, and β) responsible for different interactions were estimated. These are values that can be a design guideline for the path structure in solid media.

We believe that the series of researches introduced here points to the direction of a new development of battery materials and systems aimed at improving the battery power performance.

## 2. Evaluation Approach of Ionic Mobility and Microviscosity

Dynamic values that are the essence of evaluations here were estimated based on the theoretical model using the measured diffusion coefficients and ionic conductivities.

### 2.1. Fundamental Ways of Diffusion Coefficient and Conductivity Measurements

Diffusion coefficients *D*_Li_, *D*_F,_ and *D*_H_ were measured at 25 °C for the probed nuclear species, ^7^Li (116.8 MHz), ^19^F (282.7 MHz), and ^1^H (300.5 MHz), respectively, using the pulsed gradient spin-echo (PGSE) NMR technique with a JNM-ECP300W wide-bore spectrometer [8]. The Hahn-echo pulse sequence was used for measurements. The half-sine-shaped gradient pulse was applied twice in the sequence after the 90° and 180° pulses to detect attenuation of echo intensity according to the diffusion migration of the probed nuclear species (Figure 1) [9,10]. The typical values of the parameters for the pulse sequence are *g* = 2–4 T/m for the strength of the gradient pulse, *δ* = 0–5 ms for the gradient pulse width, and Δ = 20–50 ms for the diffusion time corresponding to the interval between the two gradient pulses. For example, the Hahn-echo pulse was detected by changing the gradient pulse width from 0 to 5 ms with the interval of 0.2 ms, and a *D* value (*D*_Li_, *D*_F,_ or *D*_H_) was estimated by fitting the measured echo intensity change to
*M* = *M*_0_ exp[−*γ*^2^*Dδ*^2^*g*^2^(4Δ − *δ*)/π^2^](1)
where *M* is the echo intensity and *γ* is the gyromagnetic ratio of each nuclear species [11].

The spin-spin relaxation time, *T*_2_, was measured using the Carr–Purcell–Meiboom–Gill (CPMG) method of NMR, and a *T*_2_ value (*T*_2_ (Li), *T*_2_ (F), or *T*_2_ (H)) was estimated from the slope of the log plots of the echo intensity against time as [12,13]
*M* = *M*_0_ exp[−*t*/*T*_2_](2)

The ionic conductivity of the gel electrolyte or the solution in a separator membrane was measured by the complex impedance method using a frequency analyzer (Model 1250, Solartron Analytical, Wokingham, U.K.) combined with a potentiostat (Model 1287) or the impedance analyzer (HZ-Pro S4, Hokuto Denko Corp. Tokyo, Japan). A constant alternating current (ac) voltage in the range of 20–100 mV was applied in the frequency range of 1 mHz to 65 kHz or to 1 MHz to detect the impedance data [14,15]. By fitting the data to the equivalent circuit model, the impedance values of gel electrolytes and an electrolyte solution within a separator membrane were estimated.

### 2.2. Identification of Ionic Mobility and Microviscosities as its Elements

In the case of the lithium electrolyte solution in which a lithium salt, e.g., LiPF_6,_ is dissolved in a binary solvent, ethylene carbonate/diethyl carbonate (EC/DEC), the lithium salt dissociates due to the solvation effect on Li^+^ and achieves the equilibrium state as
LiPF_6_ + nEC ⇆ Li(EC)_n_^+^ + PF_6_^−^(3)

Solvation structure and number (n) depends on the mixing ratio of EC to DEC and the salt concentration of the solution. Under this dissociation equilibrium, the degree of salt dissociation (*x*) is about 0.6 for 1 M electrolyte solution. At the diffusion measurement using the PGSE-NMR technique, it takes around 50 to 100 ms for diffusion time to detect an echo peak intensity change with each variable parameter, δ. Due to the fast exchange reaction between left and right sides of Equation (3) in the equilibrium state, dissociated ions and associated ion pairs cannot be distinguished normally during a diffusion time, and the obtained NMR peak reflects the mean values of ions and ion-pair, which bears the weight of dissociation degree of the salt. As a result, observed diffusion coefficients, *D*_Li_ and *D*_F_ from a series of peak attenuation with δ probed by the respective nuclear species, ^7^Li and ^19^F can be expressed as [16,17,18]
*D*_Li_ = *xD*_cation_ + (1−*x*) *D*_pair_
*D*_F_ = *xD*_anion_ + (1−*x*) *D*_pair_(4)
where *x* is the dissociation degree of the salt, and *D*_cation_, *D*_anion,_ and *D*_pair_ are the inherent diffusion coefficients of the cation [Li(EC)_n_^+^], anion (PF_6_^−^), and ion-pair (LiPF_6_), respectively. It should be noted that *D*_cation_ and *D*_anion_ directly reflect the ionic mobilities of the cation and anion.

The non-polar solvent DEC species in the binary solvent generally contributes little to cation solvation and exists independently in the electrolyte solution. Therefore, *D*_H_, of the solvent species probed by ^1^H, is obtained from the peak attenuation of DEC as
*D*_H_ = *D*_solv_ = *D*_DEC_(5)

In order to find *D*_cation_, *D*_anion_, *D*_pair_, and *x* individually, two more independent relations are needed. we have to add another expression, The Nernst-Einstein equation which represents the relation of the conductivity, *σ* as
*σ* = *σ*_cation_ + *σ*_anion_ = (*e*^2^/*kT*) *x N*(*D*_cation_ + *D*_anion_)(6)
where *e* is the proton charge, *k* is the Boltzmann constant, *T* is the absolute temperature, and *N* is the prepared carrier concentration per unit volume. In the case of a free electrolyte solution, *N* is the salt concentration of the solution, and in the case of the solution in a separator membrane, *N* is the salt content in a unit volume of the membrane [19]. *σ* can be measured by the electrochemical impedance technique. The neutral species, ion-pair, and solvent species are isolated, and they alone follow the Stokes–Einstein relation. Therefore, the diffusion coefficients, *D*_pair_ and *D*_DEC,_ and the radii, r_pair_ and r_DEC_ of those species are in the relation of
*D*_pair_/*D*_DEC_ = r_DEC_/r_pair_(7)
where r_DEC_ and r_pair_ are estimated as 2.67 and 4.9 Å, respectively, according to the van der Waals size of the atomic species [20,21]. As a result, we can estimate *D*_cation_, *D*_anion_, *D*_pair_, and *x* by solving Equations (4), (6) and (7) using the measured *D*_Li_, *D*_F_, *D*_H,_ and *σ*.

As explained above, separator membranes and polymer chain networks of gel electrolytes play the part of holding the electrolyte solutions inside with providing ion migration pathways to carry ions smoothly from an electrode to the counter electrode in order to perform charge transfer reactions. The ions, originally present in an electrolyte solution, interact with the surrounding dissolved ions and liquid solvent species first and then interact with the solid media (separator and polymer chains in the gel electrolyte) around the electrolyte solution while repeatedly colliding with each other. The type and magnitude of the interactions depend on the chemical property, size, and morphology of carrier ions, liquid solvent, and solid media.

When a macroscopic sphere experiences viscous or drag in an incompressible medium, Stokes showed that the drag force *F* opposing the sphere is given by the relation as Stokes’ low as
*F* = 6 π r η ν(8)
where *F* is the force on a carrier ion, r is the radius of the ion, η is the bulk average viscosity of the solution, and ν is the velocity of the macroscopic body [22]. Application of this relation to the real dynamics of ions in solvent species would be a bold challenge owing to the main reasons to hesitate: (1) the solvent surrounding the carrier ions is composed of the spherical or elliptical molecular species of approximately the same size as of the target ions and is not an incompressible medium, (2) the lithium cation has a solvated structure in which several solvent molecules are coordinated with repeating the exchange reaction with free solvent species, and (3) individual ion moves by sensing the local viscosity generated by the interaction with surrounding ions, molecules, and solid media. This indicates that we cannot use the bulk average viscosity in Equation (8), and the local viscosity that each ion senses is different depending on the size and the type of charge of the target ion and the surrounding species.

When a particle is engaged in its random walk, it is subject to the viscous drag force exerted by its environment. When diffusion is occurring, there is a driving force −*dμ/dx* operating on the particles. This driving force produces a steady-state diffusion flux corresponding to a drift velocity ν_d_ for the diffusing particles. The diffusional driving force −*dμ/dx* must be opposed by an equal resistive force, which will be taken to be the Stokes viscous force, 6 π r η ν_d_ as

−dμ/dx = 6 π r η νd (9)

Absolute mobility, ${\bar{\mu }}_{abs}$ for the diffusive particle is defined by dividing ν_d_ by the diffusional driving force, that is, the Stokes viscous force,
(10)$${\bar{\mu }}_{abs}=\frac{\nu_{d}}{-\frac{d\mu }{dx}}=\frac{\nu_{d}}{6\pi r\eta \nu_{d}}=1/6\pi r\eta \;$$

The fundamental expression relating the diffusion coefficient and the absolute mobility can be written from the Einstein relation as [23]
(11)$${\bar{\mu }}_{abs}=\frac{D}{kT}\;\;$$

Then the Stokes–Einstein relation is obtained as
(12)$$D=\frac{kT}{6\pi r\eta }\;\;$$

This links the process of diffusion and viscous flow.

On the other hand, the effect of ionic charge on the ionic mobility is expressed under the drift velocities in a unit electric field (1 V cm^−1^) as conventional (electrochemical) mobilities with the symbol *μ*_conv_ as [22]
(13)$$\mu_{conv}=\frac{z_{i}e_{0}}{6\pi r\eta }\;\;$$

This equation, however, does not include the significant effect of the Coulombic ion-ion interaction. In addition, *μ*_conv_ increases with increasing the charge amount, which is contrary to the general experience that the higher the charge amount, the lower the ionic mobility. It is reasonable to think that the ion-ion interaction is a component of η according to Equation (8). This is because η is not an averaged bulk viscosity but is accepted as a local viscosity each ion senses individually depending on their feature and the surrounding situation. Furthermore, not only the Coulombic interaction between the ions but other kinds of interactions on each ion could be present, that is, van der Waals interaction between the ion and the surrounding ionic and molecular species, and the specific interactions between the ion and solid media such as physical obstacles and some charged sites on the solid media. As a result, η is acceptable to be composed of three main elements as
η ⇒ η + α + β_ca_ (β_an_)(14)
where η on the right side is the microviscosity attributed to the van der Waals interaction of the target species and all surrounding species, α is the microviscosity due to the Coulombic interaction between the ions, and β_ca_ (β_an_) is the microviscosity reflecting the interaction between the cation (anion) and solid media [18,19]. Neutral species such as solvent species, EC and DEC molecules, and the associated ion pairs, e.g., LiPF_6,_ are not influenced by the Coulombic interaction and only η contributes to the total viscosity of the species in the migration process. Ionic species are influenced by both η and α. Although the Coulombic effects act among any charged species, attractive forces between the nearest cations and anions would be dominant for α because the Coulombic force is inversely proportional to the square of the interion distance. In addition, β_ca_ (β_an_) represents the microviscosity attributed to the interaction between the cation (anion) and solid media, such as the pathway wall of a separator membrane and polymer chains in polymer gel electrolytes. This value is a component of microviscosity that causes the mobility of cations or anions to be selectively affected by the presence of a solid medium. In the actual sample, it is considered that the cation and anion may have unique β_ca_ and β_an_ simultaneously. However, since it is necessary to find the solution of the simultaneous equations, this equation includes only one of which gives a positive solution. Therefore, when β_ca_ and β_an_ compete with each other in a sample, the difference between them could be estimated as a net effect. It is normal that the effect of β_ca_ (β_an_) is responsible for the Coulombic force, which is comparable to α.

As a result, the microviscosity of ions moving in a solid medium could be composed of η + α + β_ca_ (β_an_). Figure 2 shows the schematic diagram of interactions on the ions of the solution in the space of the separator membrane.

Then, the diffusion coefficients of the solvent and ionic species could be represented as follows.
$$D_{\mathrm{solv}}=\frac{kT}{6\pi r_{\mathrm{DEC}}\eta }$$
$$D_{\mathrm{an}}=\frac{kT}{6\pi r_{\mathrm{anion}}\eta^{\prime }},\;\eta^{\prime }=\eta +\alpha ,$$
(15)$$D_{\mathrm{ca}}=\frac{kT}{6\pi r_{\mathrm{cation}}\eta^{\prime \prime }\;},\;\eta^{\prime \prime }=\eta +\frac{r_{\mathrm{anion}}}{r_{\mathrm{cation}}}\alpha +\beta_{\mathrm{ca}}$$

The prefactor, *r*_anion_/*r*_cation_ on α of *D*_cation_, means that the viscosity is proportional to *F*/*r*_ion_ (*F*: Coulombic force between the cation and anion) according to the Stokes equation. Notably, when examining the interaction between the attractive sites and anion, β_an_ had to be included in the equation for *D*_an_, whereas β_ca_ had to be removed from the equation for *D*_ca_ (Equation (15). This is because our model of solving three simultaneous equations could independently estimate only three parameters (η, α, and β_ca_ or β_an_) as noted above. Then, using the previously estimated values, *D*_solv_(= *D*_DEC_), *D*_ca_, and *D*_an_, we could determine the values, η, α, and β_ca_ (or β_an_) by solving the simultaneous equations of Equation (15).

## 3. Evaluation and Discussion of Practical Materials

### 3.1. Dynamic Values and Properties of Free Electrolyte Solutions

#### 3.1.1. Solvent Dependence of Ionic Mobility and Microviscosity

Ion concentration of an electrolyte solution (*c*) is obtained from the product of the concentration of the dissolved salt (*N*) and the degree of dissociation thereof (*x*). It is acceptable that the dissociation degree of the salt increases with an increasing dielectric constant (*ε*) of the solvent due to the polar solvent species promote salt dissociation. On the other hand, from a macroscopic point of view, the ion mobility is proportional to the reciprocal of the bulk viscosity (*η*) of the solution according to the Stokes–Einstein equation. Therefore, high ionic conductivity is achieved by controlling the dielectric constant (*ε*) and bulk viscosity (*η*_m_) of the solution to achieve high ionic conductivity (*σ*), which is a product of the concentration (*c*) and mobility (μ) of ions.

The two parameters, *ε* and *η*_m,_ can be generally controlled by changing the mixing ratio of the binary solvent composed of the polar and non-polar solvents. For example, in the case of the solution, 1 M LiTFSI/EC+DMC, with increasing the fraction of high-polar and high-viscous EC solvent, *ε* and *η* of the solution increase, and with increasing the fraction of low-polar and low-viscous DMC solvent, *ε* and *η*_m_ decrease. We evaluated the changes in microscopic dynamic values, *D*_cation_, *D*_anion_, *x*, η, and α with changing the macroscopic *ε* and *η*_m_ to see the real situation of interactions between the species in 1 M LiTFSI/EC+DMC as in Figure 3 [17]. The numbers in the figure show the molar fraction ratio of EC:DMC. As is expected, the dissociation degree of the salt, *x* increased with increasing EC fraction and the consequent dielectric constant *ε* of the solution (Figure 3a). On the other hand, *D*_cation_ and *D*_antion_ showed the anomalous feature having a maximum at around EC:DMC = 3:7, which is deviating from the general expectation that the ionic mobility monotonically increases with decreasing the bulk viscosity (Figure 3b). The cause of the specific changes was revealed by the estimations of α and η represented in Figure 3c. η, the microviscosity attributed to the van der Waals interaction, monotonically increased with increasing *ε* and *η*_m_. This indicates that the bulk viscosity mainly reflects the microviscosity attributed to the van de Waals forces acting between all ions and molecular species. On the other hand, α, the microviscosity attributed to the cation-anion Coulombic interaction, showed a change with a minimum at (3:7)-solution. That is, α component in the microviscosity has an inverse relationship with ionic mobilities.

We can see that the changes in *D*_cation_ and *D*_anion_ of the electrolyte solution strongly reflect the change in α.

According to Coulomb’s law, the Coulombic force acting on ions in a solvent decreases as the dielectric constant (*ε*) of the solvent increases [25]. This could be interpreted as a shielding effect of the polar solvent species on the ions in the solvent. The decreasing manner of α with *ε* in Figure 3c in lower EC fraction range follows the general theory. However, the increase in α after the minimum point is contrary to the general expectation. It can be assumed that the anomalous increase in α with solvent polarity is associated with the solvation structure of lithium cations.

Considering the stoichiometric ratio of lithium and EC solvent species in each solution, the number of solvating EC species on a Li^+^ would increase from 0 to 3 in the range from (0:10)- to (3:7)-solution. In (3:7)-solution in which [EC]/[Li] = ~3.1, most of all the EC species are coordinated to Li^+^ to form the stable structure of the solvated lithium, Li(EC)_3_^+^ [26,27]. This cation would have a long lifetime because there are few residual EC species to exchange with the coordinating EC in (3:7)-solution in the equilibrium state. EC species stably coordinated to Li^+^ are strong barriers to the approach of TFSI^-^ to the cations, resulting in the smaller α. On the other hand, even in the case of (7:3)-solution in which [EC]/[Li] = ~7.8, the direct coordination number of EC on a Li^+^ would be 3–4, almost the same as that of (3:7)-solution, considering the same size of *r*_cation_ estimated in Figure 3a. However, the solvated lithium, Li(EC)_3_^+^ and Li(EC)_4_^+^, are further surrounded by the free EC species in the EC-rich solvent, and the exchange between the coordinating EC and free EC species would occur fast in the equilibrium state of the solution. The frequent exchange reactions would also promote the movement of all species in the solution and provide many chances for the TFSI^−^ anions, as for the free EC species, to approach the lithium cations. As a result, the TFSI^−^ anions in the EC-rich solvents would be, on average, closer to the lithium cation compared with that of (3:7)-solution with little movement of the species, showing larger α. On the contrary, in the case of the EC-poor solutions such as (1:9)-solution, TFSI^−^ anions could stand, on average, near the lithium cation due to the lower dissociation degree of the salt. Therefore, α would again be larger than that of (3:7)-solution.

#### 3.1.2. Comparison of Two Solutions with Different Solvation Structures of Li^+^

The temperature dependence of dynamic values of two typical electrolyte solutions, L1; 1 M LiPF_6_ /EC+DEC (EC:DEC = 1:1) and L3; 1 M LiPF_6_ /EC+DEC (EC:DEC = 1:6.9) was estimated in Figure 4 [27]. These electrolytes, different from LiTFSI /EC+DMC of Figure 3, were selected due to the fact that L1 electrolyte is widely used in practical batteries and can be useful reference data. Considering the different molar fractions of the solvent species against Li (Li:EC:DEC = 1:7.2:4.1 for L1 and 1:1.8:6.9 for L3), and morphological stability of solvation structures coordinated in a triangle and tetrahedral, Li(EC)_4_^+^ would be the dominant solvated cation in L1, while Li(EC)(DEC)_3_^+^ and Li(EC)_2_(DEC)_2_^+^ would be preferred in L3 due to lack of EC species [28,29]. Solvating EC displays stronger coordination than that of DEC due to the higher polarity of EC molecules. As a result, lithium cations of L1 have a strong shielding effect against surrounding species containing anions. That is the reason for α(L1) < α(L3) of Figure 4b. On the other hand, it was found that η(L1) > η(L3), which is correlated with the magnitude of macroviscosity, η_m_ of the solvents, η_m_(EC) > η_m_(DEC). Based on these results of η and α of Figure 4b, we could recognize that the orders of *D_ca_*(L1) > *D*_ca_(L3) and *D*_an_(L1) > *D*_an_(L3) in higher temperature regions (Figure 4a) are dominated by α(L1) < α(L3).

However, it is noted that *D*_ca_(L1) < *D*_ca_(L3) at the temperature range below 20 °C opposite to the order at the higher temperature range. This reflects the anomalous change in r_ca_ (Figure 4c), estimated from Equation (15), of L1. Since the mobility of species decreases in the low-temperature range, it becomes difficult to exchange EC of the solvate ligand, and the free EC species are gathered around the solvate cation Li(EC)_4_^+^ with forming the second and third solvated layer considering the rapid increase in r_ca_ with decreasing temperature. Along with this, the apparent cation size (r_ca_) increases, and as a result, the cation mobility (*D*_ca_) also decreases.

### 3.2. Dynamic Values of Electrolyte Solution in Solid Media

As is said first in the Introduction section, the causes of interaction between the target ions and surrounding species in the solvent or between the target ions and solid media are roughly categorized into two types, physical obstacle effect and chemical attractive/repulsive effect. The physical effect depends on the shape and morphology of the obstacles that affect the collision and subsequent diffusion behavior of the ions, and the chemical effect depends on the number and strength of the interactive species in the solvent and sites on the solid media. Table 1 summarizes the overall picture of interactions between the ions and solid media that we have studied so far. Physical factors of separator membranes responsible for the interactions on the moving ions are porosity, pathway width, pathway tortuosity, and cross-sectional shape of the pathway. Moreover, chemical factors of separator membranes concerning the interactions on the ions are polar groups and some adsorbed species on the inside wall of pathways. On the other hand, physical factors of polymer gel electrolytes on the interactions are the crystalline size of the polymer prepared in the gelation process, and the chemical factors are polar groups such as -OH groups, ether -O- groups originally present, and Lewis acidic anion groups artificially adsorbed on the polymer chains. The specific contents of each are as follows.

#### 3.2.1. Physical Properties of Separator Membranes

There are four main morphological factors of separator membranes that affect the ion migration in the separator; membrane porosity and pore size in the membrane, tortuosity, and cross-sectional shape of pathways composed of linked pores in the membrane. The pore spaces in a membrane are prepared by membrane stretching in the uniaxial or biaxial direction in the membrane plane. The porosity of a membrane is the ratio of the internal cavity volume to the outer volume of the membrane. Porosity and pore size, which correspond to the number of paths and their width for ion transport, depend on the stretching conditions for the preparation of porous membranes. Tortuosity, an index reflecting the degree of curve of pathways, is generally defined as a ratio of the mean effective migration distance of the ion path across the membrane surface to the separator membrane thickness. High tortuosity and lengthening of the ion migration distance lead to an increase in the electrical resistance in proportion to the migration distance even if the ionic conductivity (*σ*/Scm^−1^) is a constant independent of the path tortuosity. However, when ions migrate in a path of restricted space within a separator membrane, their mobility, μ (μ = *σ* /(*en*)) inevitably decreases because the ionic collisions with the wall of the pathway at least physically inhibit the free diffusion of ions. In other words, even though the tortuosity, as identified by the macroscopic path length and membrane width, generally represents the macroscopic and geometrical features of pathway length, it can also be recognized as a microscopic factor responsible for the mobilities of the species moving in the pathways. With increasing the tortuosity of pathways in a membrane, the contribution of the ionic collisions with the pathway wall to ionic mobility increases because of the increase in the wall area that the ions come across during the migration through the path across the membrane surface, and the increase in the number of inflection points of the path as a rate-determining site where the movement of ions is suppressed. As a result, ionic collision frequency with the pathway wall is higher in the tortuous path compared to the linear path even with the same total distance, leading to low ionic mobility.

Figure 5 represents the estimated results of ionic mobilities, *D*_cation_ and *D*_anion_, and the microviscosities, η, α, and β of the electrolyte solution, L1, which was identified in Figure 4, in polyethylene (PE) separator membranes as a function of membrane porosity [24,30]. PE membranes were prepared by the conventional wet method with stretching in biaxial direction and have the randomly arranged pathways [3,31]. When the porosity of the membrane increases, a pathway network is formed in which the paths merge in places, and as a result, it becomes possible for carrier species to select a path having low tortuosity and smooth movement. That means the tortuosity of the path that the ions actually pass would be lower than that estimated from the apparently complex path structure composed of the randomly arranged pathways.

Contrary to the monotonous increase in *D*_anion_, *D*_cation_ showed the specific change with a maximum with increasing the porosity (Figure 5a). This result reflects the sum of the changes in η, α, and β(Figure 5b,c), in which change in η was slight, and the effect on the changes in *D*_cation_ and *D*_anion_ would be negligible. α, attributed to the cation-anion interaction, steeply decreased with increasing the porosity. This is because the interaction between the cations and anions increased as a result of suppressing the movement of ions in a narrow space due to the low porosity. This is similar to the nature of ions in the concentrated solution that suppresses the free diffusion of high-density ions. As a result, α became larger, leading to the smaller *D*_cation_ and *D*_anion_ with the decrease in porosity. At the same time, β_an_ > 0 appeared in the lower range of porosity. Furthermore, with increasing the porosity, β_an_ decreased and, on the contrary, β_ca_ increased in the range over 60% of porosity. These changes of β_an_ and β_ca_ with porosity lead to the specific decrease in *D*_cation_ in the range over 55% of porosity. The change of the type of β (β_anion_ ↔ β_cation_) with porosity can be explained as follows.

As shown in Table 1, there are two types of factors for the interaction between ions and membrane, namely, a physical factor attributed to collision frequency associated with the pathway morphology, such as tortuosity, path width, and cross-sectional shape of the path, and a chemical factor between the specific ions and the effective sites of the membrane represented by Coulombic interactions. As discussed above, a decrease in the porosity of the membrane generally leads to a decrease in pore size and an increase in tortuosity of linked-pore pathways composed of a limited number of pores. This tendency is reflected in the change in β_anion_ > 0 in Figure 5c: the specific appearance of β_anion_ in the low-porosity region would be due to the frequent collision of anions with the path wall during movement in the path. It is probable that the anion with a small size and high charge density would be sensitively affected by the inflection zone of paths compared to the solvated cation, which is charge-shielded by the coordination solvent species.

On the contrary, β_cation_ > 0 appeared in the range above 57% porosity and increased with porosity. In the membranes with higher porosity, the pathways connect in places to form a pathway network so that ions can select more efficient migration paths regardless of the tortuosity estimated from the actual form of the paths. In the field, the number of times anions are trapped at inflection zones would be reduced. Further, if the path width is increased with porosity, the influence of the collision with the path wall becomes smaller, which is similar to the diffusion nature in the free solution as reflected in reduced α. It is considered that the cause of β_cation_ appearance is that the influence of the Lewis-basic sites originally existing on the path wall of PE membranes became easier to see in the higher porosity membrane in which the influence of path tortuosity is reduced.

As was explained in α change, higher porosity provides higher mobility of ions, which leads to enhanced collision frequency of the ions with the path wall and promotes the specific interaction attributed to the surface condition of the path wall. In the case where the pathway walls carry a negative electric charge, which causes ζ potential to develop on the membrane, cation species are attracted selectively by the membrane with β_ca_ > 0 [32]. This is the main reason that *D*_ca_, which first increased owing to the decreases in α and β_an_, decreased from the maximum point followed by β_ca_ increase with increasing the porosity.

When we used L3 solution (see Figure 4) in place of L1 solution in the PE membranes, β_ca_ > 0 appeared in all range of porosity of the membranes used in this research as shown in Figure 5d. This would mean that the Coulombic cation–membrane interaction in L3 surpassed the physical anion-membrane interaction, even in the low-porosity membranes. This attributes to the difference in the cation solvation structure between L1 and L3 solutions. As the cation of L3 solution has weakly coordinated DEC species compared with the cation of L1 solution strongly coordinated with EC species, the surface positive charge density could be higher than that of the cation of L1. As a result, the Coulombic interaction between the L3 cation and the path wall was dominant in the path regardless of the membrane porosity and path form.

In designing the separator membranes, it is important to control the tortuosity of ion transport pathways that are directly associated with the internal resistance and the consequent power performance of batteries. We originally evaluated the effect of path tortuosity on ionic mobility and microviscosities by preparing the specific polypropylene (PP) separator membranes with straight pathways across the membrane surface, as shown in Figure 6 [33].

PP porous membranes were prepared under dry conditions via the uniaxial stretching method by changing the applying conditions to control the degree of pore alignment [34,35]. It is acceptable that the path tortuosity of the porous membrane is reflected in the macroscopic Gurley permeability (*G*/s (100 cm^3^)^−1^), which is the time required for the air of 100 cm^3^ to pass vertically through the membrane surface. Strictly speaking, the *G* value depends on two elements, pore volume in which the introduced air can enter and the air mobility passing through the pore space [36]. Pore volume depends on the thickness and porosity of the membrane, and the mobility depends on the width and tortuosity of pathways composed of the linked pores. To extract the component corresponding to the mobility from the *G* value, *G*′ (normalized *G*) was newly defined as
*G*′ = *G* × (*d*_0_/*d*) × (*p*/*p*_0_)(16)
where *d*_0_ and *p*_0_ are the standard values of membrane width and porosity, respectively, to normalize the volume-dependent part of the *G* value. In the research, *d*_0_ = 16 μm and *p*_0_ = 55% were applied as they are averaged values of the membranes used.

Figure 7 compares the changes of ionic mobilities and microviscosities between L1 and L3 electrolyte solutions within the polypropylene (PP) membranes of straight pathways and random network pathways as functions of *G*′ and path diameter, 2*r* [33]. The plot data in the range of *G*′ < 70 corresponds to the values of the straight-pathway membranes, and the data of *G*′ > 70 was obtained from the membranes with random network pathways. The shape of the linear path was the same for all membranes regardless of the porosity of the straight-path membranes from the SEM observation. Therefore, the difference in *G*′ values of the straight-path membranes would attribute to the difference in the mobility associated with the path diameter, 2*r*. *D*_ca_ and *D*_an_ monotonically decreased with increasing *G*′ (Figure 7a). In order to clarify the causes of this change, we plotted η, α, and β_ca_ as functions of *G*′ and 2*r* (Figure 7b,c). Compared with the microviscosities of free electrolyte solution (η_0_, α_0_, β_0_, the plots at *G*′ = 0), α apparently increased and β_ca_ appeared and steeply increased with the increase in *G*′, contrary to the slight increase in η of L1 and L3 in PP separator membranes. In more detail, α value changed more slowly between the membranes than the steep increase from the free electrolyte solution to the solution within the membrane, while there was a large difference in β_ca_ value between the straight pathway membranes (β_ca_ ~ 0 = β_0_) and the random pathway membranes (β_ca_ > > 0) bordered by *G′* ~ 70. These mean that enlarged α in the membrane compared to α_0_ reflects the presence of obstacles of path wall independent of the tortuosity, and an increase in β_ca_ in membranes reflects the increased path tortuosity of the membranes.

It should be noted that β_ca_ ~ 0 in the straight-pathway membranes does not mean the ion do not collide with the wall of pathways during migration. Regardless of the magnitude of pathway tortuosity, when ions move in the restricted space of paths (2*r* ≤ 1 μm), they cannot avoid collision with the inside wall of pathways owing to their essential nature of random walk migration. However, the ionic collision frequency with the wall would be higher in the pathways of high tortuosity because of the larger number of inflection points in a path, with decelerating ion movement and reducing the mobility. Compared to the tortuous paths, the collision frequency of ions moving in the straight pathway would be lower. The fact that apparent β_ca_ was around zero for L1 within the straight-path membranes in Figure 7b indicates the ionic mobility in the straight paths was dominated by the greatly increased α reflecting the decreased motility of cations and anions in the restricted space.

It is easy to understand the effect of 2*r* on η, α, and β_ca_ (Figure 7c) if straight-path membranes and random-path membranes are discussed separately. It is noted that α has a small 2*r* dependence in each type of membrane (with straight-path and random-path) with L1 or L3 solution. On the other hand, β_ca_ of random-path membranes with L1 solution strongly depended on 2*r* although β_ca_ of L3/random-path membranes and L1 or L3/straight-path membranes showed smaller change with 2*r*. Considering these results comprehensively, β_ca_ attributes to the tortuosity of the pathway, and α is dominated by the presence of the barrier in the migration space and is not as sensitive to the changes in the pathway structure as β_ca_.

In order to evaluate the effect of pathway tortuosity on the ionic mobility and microviscosities in more detail, we measured the diffusion coefficients in different directions, along and inclined to the straight pathway composed of aligned pores, as shown in Figure 8 [37]. The stacked PP sheets were placed in a nuclear magnetic resonance sample tube (ϕ = 5 mm) so that the planes of the films were (1) vertical and (2) inclined at an angle of 53° from the longitudinal axis of the tube. As the diffusion measurement is performed in the z-direction along the tube length for both samples, the two film configurations permitted the diffusion measurement in different directions in the path: (1) along the length of the path for the vertical samples and (2) at an inclination to the length of the path for the inclined samples of the aligned-pore membranes.

The diffusivities along and inclined to the pathways correspond to the migrations in paths of tortuosity 1 and 1/sin 53° = 1.25, respectively, based on the macroscopic definition of tortuosity.

Contrary to the conductivity parallel to the straight pathway of the vertically set membranes, we could not directly measure the conductivity corresponding to diffusion at an inclination *θ* to the straight pathway shown in Figure 8b because of the difficulty in setting up the electrodes to detect current signals in that direction. Therefore, we obtained *σ_θ_* from *σ*_0_ using the relation
*σ_θ_* = cos^2^*θ* · *σ*_0_(17)
which is based on the assumption that *σ*_0_ is isotropic and *σ_θ_* corresponds to the component of *σ*_0_ at an inclination of *θ*. If the ionic collisions with the wall make larger contributions that reduce the ionic mobility, *σ_θ_* would be anisotropic because of the anisotropic ionic mobility, depending on the diffusion direction with respect to the wall, and the actual value of *σ_θ_* would be smaller than that obtained from Equation (17); i.e., *σ_θ_* < cos^2^*θ* · *σ*_0_. With decreasing *σ_θ_* in response to the anisotropy of mobility, the calculated values of *D**^θ^*_ca_ and *D**^θ^*_an_ decrease and subsequent η*^θ^*, α*^θ^*, and β*^θ^* increase because the decreased ionic mobilities reflect enhanced interactions between the ions and the surrounding species or membrane. Therefore, η*^θ^*, α*^θ^*, and β*^θ^* obtained based on the assumption of Equation (17) would be minimum expected values.

Figure 9 shows the estimated values of the ionic mobilities of the electrolyte solution in the vertical separator membranes (*D*^0^_ca_ and *D*^0^_an_) and inclined separator membranes (*D**^θ^*_ca_ and *D**^θ^*_an_) [37].

*D**^θ^*_ca_ and *D**^θ^*_an_ of the inclined membranes were lower than the respective *D*^0^_ca_ and *D*^0^_an_ of the vertically set membranes, except those with random paths (*G*′ > 100) that showed *D*^0^_ca_ ≈ *D**^θ^*_ca_ and *D*^0^_an_ ≈ *D**^θ^*_an_ (comparison of Figure 9b and Figure 9a). For the membranes with straight paths, *D*^0^_ca_, *D*^0^_an,_ and *D**^θ^*_an_ decreased with *G*′ and increased with 2*r* as expected, contrary to no noticeable change in *D**^θ^*_ca_ with *G*′ and 2*r*. These features indicate that cation mobility depends on the diffusion direction in the straight pathway; i.e., the cation mobility is more strongly affected by ionic collisions with the pathway wall than the anion mobility. In order to clarify the cause of these results, we estimated the microviscosities of the ions as in Figure 10 of the two types of membranes, vertical membranes (identified by superscripts 0) and inclined membranes (identified by superscripts *θ*) [37].

First, it is apparent that the random-path membranes showed η^0^ ≈ η*^θ^*, α^0^ ≈ α*^θ^*, and β_ca_^0^ ≈ β_ca_*^θ^* in contrast to η^0^ ≲ η*^θ^*, α^0^ < α*^θ^*, and β_ca_^0^ << β_ca_*^θ^* of the straight-path membranes (comparison of Figure 10a and Figure 10b). This indicates that the random-path network is isotropic, and the ion diffusivity in it is independent of the diffusion direction in a membrane. The differences in η, α, and β_ca_ between the vertical and inclined membranes are apparent by the comparison of the plot against 2*r* for the membranes with straight pathways (Figure 10c,d). The most noticeable difference between them is the enlarged β*^θ^*_ca_, which comes from the increased collision frequency in the diffusion in the inclined direction as shown in the lower of Figure 8.

Another significant point of Figure 10 is that η*^θ^*, α*^θ^*, β*^θ^*_ca_ increased steeply with decreasing 2*r* below 0.3 μm (Figure 10d). This tendency is reflected in the change of α^0^ of the vertical membranes to some extent. These behaviors indicate that the effects of collision (β_ca_) and the reduced mobility (α) in the restricted space on the ionic mobility in it increase sharply when the path width is 0.3 µm or less. In the path of width more than 0.3 μm, the values were almost independent of 2*r*, although the restrictions on movement within the paths remain severe compared with the values of free electrolyte solution (μ_free_ < μ^0^, α_free_ < α^0^, β_ca,free_ = 0).

It is also expected that the interaction between the carrier ions and path wall of separator membranes depends not only on the path tortuosity, which is a parameter of path form along the path length but on the cross-sectional shape of the pathway, which is a parameter of the distance between the moving ion in a path and the path wall across the path length. The cross-sectional shape of the pathway of PE separator membranes depends on the stretching ratio between the machine direction (MD) and transverse direction (TD) in the formation process of porous membranes. A large difference in the ratio led to the anisotropic cross-sectional shape of pathways, which was characterized by anisotropy (*f*_(TD/MD)_) of the elastic moduli in MD and TD (*E*_MD_ and *E*_TD_), which were determined by the standard test method [38], as well as the separation of mode pore size (2*r*_max_) and median pore size (2*r*_med_) determined by pore-size distribution using the mercury intrusion porosimetry method. Especially, 2*r*_max_ against porosity of the membranes divided into three separate linear correlations, and each group corresponded to the respective range of *f*_(TD/MD)_, ~0.1, 0.3‒0.8, and ~0.9 as shown in Figure 11 [15]. These results suggest that the membranes of C and D groups, which include the membranes prepared with the respective stretching ratios of MD = 9.0, TD = 5.0 and MD = 9.0, TD = 9.0, with largest 2*r*_max_ and *f*_(TD/MD),_ have the most anisotropic cross-sectional shape of the pathways. In practice, in the region where *f*_(TD/MD)_ is 0.8 or more, *D*_ca_ and *D*_an_ decreased drastically correlated with the increase in α attributed to the cation-anion interactions, and β_ca_ attributed to the cation–membrane interaction in the pathways of PE membranes as shown in Figure 12 [15].

#### 3.2.2. Chemical Properties of Separator Membranes

PE and PP membranes which have been generally used as separators, are non-polar materials. Therefore, it is generally expected that there is no chemical or Coulombic interaction between the ions and membrane substrate when ions move through the paths of the separators. However, in practice, we found β_an_ > 0, reflecting the anion‒membrane interaction, and β_ca_ > 0 responsible for the cation–membrane interaction, depending on the membrane porosity and solvation structure of lithium cations as carriers in PE membranes (Figure 5), and β_ca_ > 0 responsible for the cation‒membrane interaction in PP membranes (Figure 7). As discussed in Section 3.2.1, β_an_ > 0 in PE membranes was caused by the physical barrier effects of wall and inflection points of pathways. As a result, it could be assumed that β_ca_ > 0 is derived from the essential chemical feature of the PE membranes. On the contrary, β_an_ > 0 of the PP membranes independent of the width and tortuosity of the path would also attribute to the chemical feature of the PP membranes. Considering the low polarity structures of PE and PP, it is elucidated that those chemical interactions attribute to the sites acquired in the membrane formation process and are associated with the surface conditions of the inside wall of pores in the membranes. Then, we would like to verify the manufacturing process of PE and PP separators first.

PE separator membranes were prepared based on the conventional wet process as follows [3,15,31]. PE powder and liquid paraffin were mixed and heated to promote dissolution. The solution was pushed and cooled to form a base tape of PE. The tape was extended in the vertical and horizontal directions and immersed into methylene chloride to extract liquid paraffin for forming the porous membranes. The membrane porosity depends on the amount of liquid paraffin relative to PE powder at the mixing process, and the size and orientation of pores depend on the stretching strength and stretching ratio of vertical to horizontal directions. On the other hand, PP separator membranes were prepared under dry conditions via a uniaxial stretching method by changing the stretching conditions such as the repeating number of extensions, strength, and temperature to control the pore size, porosity, and, further, degree of pore alignment [34,35]. Therefore, during the pore formation process, it is possible that the inside wall of pores could be charged up due to the application of pressure and the structural change at the inner surface of pores. Especially in the case of PE membranes, the inside wall of pores is more amorphous than the bulk of the membrane because it was in direct contact with paraffin prior to extracting paraffin. It is expected that the random network of polymer chains at the inner surface of pores could attract static charge easily.

#### 3.2.3. Physical Properties of PVDF-Based Polymer Gel Electrolytes

Polyvinylidene Difluoride (PVDF)-based polymer gel electrolytes are known as physically cross-linked gels in which PVDF amorphous phase, including the electrolyte solution, is maintained inside the crystalline network, forming a swollen gel [39,40,41]. As the cross-links composed of the aggregated crystallites are weaker than the covalent cross-links as of poly(ethylene oxide) (PEO)-based gels, resulting in lower elasticity of the polymer network, the PVDF swollen gel is not adequate to retain the solution inside stably. However, we found that the ion dynamic feature of PVDF-based polymer gel electrolytes significantly depends on the morphological features of the coexisting crystalline and amorphous phases. This is an effective finding that further expands the application area of PVDF membranes.

The PVDF gel electrolyte prepared by quenching process showed a significantly short relaxation time of the anion, *T*_2_ (F), compared with that of the gel prepared by the annealing process. This is in contrast to almost the same values *T*_2_ (Li), the relaxation time of the cation, between the quenched and annealed gels, as shown in Figure 13 [42]. This indicates the anion species in the quenched gel are located where they are especially interacted associated with the crystal morphology of the quenched gel.

In the cooling process of a PVDF solution for gelation, PVDF crystalline nuclei first precipitate. The precipitated crystallites grow as crystals with forming crystal domains by drawing in the neighboring polymer chains as the gelation progresses. Then, the crystal domains combine with surrounding amorphous polymer chains that include the solution to form a gel network structure swollen with the electrolyte solution at the equilibrium state. The apparent difference between the quenched and annealed gels formed by such a process is the domain sizes of the crystalline and amorphous phases associated with the difference in length of gelation time.

In the annealing process, initially precipitated crystallites grow while incorporating the neighboring polymer chains to form large crystal domains. With increasing the crystalline domain size, the sizes of the amorphous domains, including the solution, and the puddles of the solution that could not be incorporated into the amorphous phase would increase to fill the space among the large crystal domains. As a result, the annealed gels are composed of a mixture of larger-sized crystalline and amorphous domains, as shown in Figure 14b [14]. On the contrary, there is no time for the precipitated crystallites to grow by the incorporation of surrounding polymer chains for the gels prepared by the quenching process. Then the smaller-sized crystal domains and the consequent smaller-sized amorphous domains coexist in the quenched gel as in Figure 14a. As a result, it is expected that the contact interface between the amorphous and the crystal domains is wider in the quenching gels of the mixture of finer crystal and amorphous domains. We can assume that the anomalously shorter *T*_2_ (F), which reflects the selective restriction in anionic motion due to the special interaction with the surroundings, of the quenched gel attributes to the wider interface between the crystal phase with some interactive site and amorphous phase including the anion species caused by the smaller size of the constituting domains. This was confirmed from the presence of microviscosity attributed to the anion‒polymer interaction β_an_ > 0 of the quenched gels contrary to β_ca_ > 0 of the annealed gels, as shown in Figure 15d [42]. The appearance of β_ca_ > 0 of the annealed gels could be interpreted as the effect of the cation‒polymer interaction in the large amorphous domains.

Most of the anions in the amorphous domains are located far from the crystal domain surface due to the large-sized amorphous domains and are unaffected by the crystal phase. Then, the situation of β_an_ ~ 0 in the annealed gel represents the presence of intrinsic β_ca_ > 0 that may attribute to the essential interaction between the cation and the PVDF polymer with polarity. The great difference between β_an_ > 0 of quenched gel and β_ca_ > 0 of annealed gel at 15 wt% of PVDF is reflected in the enhanced cation transport number (*t*_Li_) of quenched gel compared with the reduced *t*_Li_ of the annealed gel as expected from Figure 15b.

The difference in PVDF gel morphology attributes not only to the cooling condition in gel formation but to the crystallinity of the constituent PVDF polymer before gelation. *T*_2_ (F) of the annealed gels significantly increased with increasing the crystallinity of the polymer (*X*_c_), contrary to the *X*_c_ independent features of *T*_2_ (Li) of the annealed gels and *T*_2_ (Li) and *T*_2_ (F) of the quenched gels as shown in Figure 16 [14]. This implies that the gel morphology obtained by annealing depends sensitively on the crystallinity of the constituent polymer.

As explained above, the annealing process could promote crystal growth during the gradual change in temperature for gelation. In the gelation process, it is acceptable that high-crystallinity polymers can easily form large crystals and crystal domains because they inherently have a high degree of orientation even after dissolved in a solvent. Therefore, the gels with larger crystal and amorphous domains have longer *T*_2_ (F) due to larger amount of free anion species apart from the influence of the crystalline phase, and the gels with smaller crystalline and amorphous domains have shorter *T*_2_ (F) due to short distance and a wider interface between the crystal and amorphous domains, which promotes the interaction leading to the restriction of anionic motion. This assumption of the crystallinity effect on the gel morphology was reflected in the change in β values with *X*_c_ of the constituent polymer as shown in Figure 17 [14]. In the case of annealed gels, β_an_ appeared in lower *X*_c_, decreased, and changed to β_ca_ with increasing *X*_c_ (Figure 17d). In the case of low crystalline PVDF polymer, compared with highly crystalline PVDF, the polymer crystal is less likely to grow even in the gels prepared by the annealing process. Therefore, the number of anions that interact with the crystallites phase is relatively high even in the gel with the low crystalline PVDF, and the anion mobility is restricted. On the other hand, the quenching process for gelation prepared smaller crystalline and amorphous domains even with the use of high-crystallinity PVDF, as shown in Figure 17c.

Further, it should be noted that α < β_an_ in the quenched gels and α > β_an_ (β_ca_) in the annealed gels (Figure 17c,d). In particular, the significant difference in α value between the annealed and quench gels would be caused by the presence of β_an_ in the quench gels. This is also evident in Figure 15c,d. This means that the degree of freedom of the cation and its high mobility supported by smaller α can be effectively enhanced by suppressing the movement of the anion revealed by larger β_an_.

#### 3.2.4. Chemical Properties of PEO-Based and PVB-Based Polymer Gel Electrolytes

The polar groups on the polymer chains can selectively interact with the ions depending on the type of charge and control the specific ionic mobility in polymer gel electrolytes. In the gel electrolytes based on poly(ethylene oxide) (PEO) and poly(vinyl butyral) (PVB) [43], the respective Lewis-basic polar groups, ether oxygen (-O-) and hydroxyl group (-OH), become the cross-linking points of a polymer network with promoting the salt dissociation of the electrolyte solution by attracting the cation species via Coulombic interactions. However, the interactions of Lewis-basic polar sites generally accelerate the decrease in cation mobility, as shown in Figure 18 [17], which is a disadvantage for electrolytes of lithium batteries.

We then tried to enhance the cation mobility (*D*_ca_) by adding the Lewis acidic ionic groups on the PVB-based polymer chains, as in Figure 19a [18]. By using the ionic PVB polymer for gelation, *D*_ca_ increased as well as the decrease in *D*_an_ compared to the values of the gel composed of the original PVB without the ionic groups (Figure 19b). It is found the changes in *D*_ca_ and *D*_an_ attribute to the occurrence of the interaction between cation‒anion interaction, as shown in Figure 19c. This feature is also supported by the solvation condition of the lithium cation in the gel. The solvating layer of EC around the Li^+^ would be a barrier, suppressing the Coulombic interactions from the TFSI^-^ anion (responsible for α) and OH^−^ site (responsible for β_ca_) on the Li^+^. The larger the number of the solvating EC species, the stronger the shielding effects against the Coulombic interactions of the polar groups and anion become. This result suggests that simultaneous control of two elements, the interaction between ions and polar groups, and the solvation structure of the cation that promotes the interaction, is effective in controlling the ion mobility rationally.

#### 3.2.5. Effect of Solvation Structure of Li^+^ on Ion Migration

As mentioned above, the solvation structure of Li^+^ cation is a significant factor that affects the migration features and mobilities of cations and anions located with solid media. This is represented by the typical restricted diffusion manner, which is different from the general free diffusion according to the random walk migration of lithium species in separator membranes. In order to control the efficient ion movement in battery systems, it is indispensable to investigate the causes of the restricted diffusion clearly. Here, I will discuss the specific phenomena and their causes of restricted diffusion behaviors.

The restricted diffusion is characterized by the anomalous oscillating feature of NMR echo attenuation in diffusion measurement in contrast to the free diffusion manner that decays linearly in the log-plot of echo intensity against δ^2^Δ of Equation (1) [44,45,46]. Free diffusion is a phenomenon wherein the species in a gas or liquid undergo random walks while colliding with surrounding species according to thermal motion [47]. In the field of free diffusion, diffusion coefficients and the consequent microviscosities of each ionic species can be estimated individually through the NMR measurement with the pulse sequence for diffusion coefficient estimation, which was explained in Section 2. On the other hand, restricted diffusion behavior indicates a regular collision of the species with the surrounding barriers or sites, which disturb the random walk movement [48,49,50]. These collisions can occur via both physical obstacles and chemical interactions, resulting in the oscillation of echo attenuations deviating from the linear decay, which reflects free diffusion. Chemical interactions usually reduce the ionic mobility due to the appearance of β_ca_ or β_an_ values, as mentioned above. However, when the interaction becomes stronger, ion transfer after the interaction could be delayed, and the diffusivity deviates from random diffusion manner. Several simulated results for restricted diffusion in the ideal restricted space, such as that between two parallel barriers, showed that the oscillation form of echo attenuation reflects the size between the barriers and the geometrical form of the restricting space [44,48].

While observing the diffusion behavior of ions in practical ionic conductors, we have often encountered the restricted diffusion behavior mainly in the presence of physical obstacles such as entangled polymer chains in inhomogeneous polymer gel electrolytes [51,52,53] and the path wall of separator membranes [15,51]. Apart from these phenomena, which are easy to predict the cause, we found another significant effect leading to the restricted diffusion manner of lithium cations in electrolyte materials in separator membranes.

In the PE separators consisting of the paths with high tortuosity, the diffusive feature of the lithium cation species showed a typical restricted behavior in contrast to the anion species that follow free diffusion in the same environment, as shown in Figure 20 [51].

As mentioned above, when an ion travels a defined path, regardless of the charge type of solvation structure, it cannot avoid physical collision with the inside wall of the path due to the original nature of random diffusion even in the restricted space. However, linear changes of echo decay responsible for the anions (^19^F) in Figure 20, different from the oscillating manner of the decays of the cations (^7^Li), suggest that the path wall itself as a physical barrier is not the main cause of the restricted diffusion behavior because *D*_ca_ and *D*_an_ of the free electrolyte solution showed the same form of linear decay. It is reasonable to think that the linear changes of anion echo decays reflect that the anions are expanding their range of movement through the tortuous paths while repeatedly colliding with the path wall as if they follow a slow random walk migration from a broad perspective.

Therefore, it is reasonable to assume that the restricted features of the cation species in Figure 20a,b attribute to the essential nature of lithium cation species. We can assume two possibilities responsible for this phenomenon: (1) the positive charge of the cations and (2) the solvation structure of the lithium cations, which are coordinated by several solvent species in the equilibrium state of the solution. To verify the possibility of (2) the cations, the echo attenuations of ^7^Li were compared between the L1 and L3 solutions with different lithium cation solvation structures, as shown in Figure 20a,b. The oscillation of the echo attenuation responsible for the restricted diffusion was clearer for the cation in L1, while the restricted diffusion behavior of the cation in L3 was weakened, which is still in contrast to the linear decays of anion species in both L1 and L3.

As discussed in Section 3.1, the main structure of solvated lithium cations in L1 would be stable tetrahedral coordination, Li(EC)_4_^+^. The solvated cations are further surrounded by free EC species in L1, and a rapid exchange occurs between the coordinated EC to Li^+^ and the free EC species in the solvent under the dissociation equilibrium of the lithium salt as Equation (3). On the other hand, the L3 solution has a reduced quantity of EC species compared to L1; therefore, the stable four-coordinate solvation structure of Li^+^ is expected to be Li(EC)_3_(DEC)^+^ and Li(EC)_2_(DEC)_2_^+^, as deduced from the stoichiometric ratio of EC to DEC of L3. As the coordination of the non-polar DEC species is weak, the solvation structures are unstable, leading to a smaller degree of salt dissociation in L3 than in L1. Furthermore, almost all EC species in the solution contribute to the solvation for Li^+^ in L3 first; therefore, there are no free EC species in L3 available to exchange with the coordinated EC, unlike in L1 with rich EC. As a result, the solvated EC to Li^+^ in L3 has a longer lifetime on the lithium cation compared with EC coordinated to Li^+^ in L1, which can be exchanged for free EC species fast and repeatedly. Furthermore, when EC exchange reactions and moving collisions of the solvated cations of L1 occur simultaneously at the surface of the path wall, it takes longer for the cations to move again after the collision, and the diffusion manner deviates from the random walk movement as the anions that collide with the path wall repeatedly without any structural change.

As an index for evaluating the effects of the solvent exchange reaction on the collisions of ions with the path wall, the expected number of exchange reactions during one collision is considered by estimating the duration of a collision based on random walk theory. When we assume one-dimensional diffusion based on the random walk model with a step time Δ*t* and a step length Δ*x*, the diffusion coefficient *D* is defined as
(18)$$D=\;\frac{{\left(\Delta x\right)}^{2}}{2\Delta t}\;\;$$

Equation (18) can also be rearranged to be understood as a function of a certain small distance, *l*, and the required time, *τ*, to move this distance by diffusion as
(19)$$\tau =\;\frac{l^{2}}{2D}\;\;$$

To consider the collision of a particle having a certain size, *r*, with the path wall, *r* can take the place of *l* in Equation (19). As a collision is composed of forward and backward movements between the bulk solution and the surface of the wall, the total time required for one collision can be estimated as
(20)$$2\tau =\;\frac{r^{2}}{D}\;\;$$

Assuming *r* ~ 5 nm, and *D* ~ 10^−10^ m^2^s^−1^, as the typical size and the diffusion coefficient of solvated lithium cations, respectively, 2*τ* ~ 3 × 10^−7^ s could be obtained, which is about two orders of magnitude longer than the time constant of the ligand exchange reaction of, for example, a hydrated lithium cation, 5 × 10^−9^ s [54]. This implies that numerous solvent exchanges occur during a single collision, and it is reasonable to assume that the collisions are affected by these solvent exchange reactions, which in turn delay such collisions and, in some cases, result in the restricted diffusion manner observed here.

Another possibility of the restricted diffusion manner is the positive charge of the cation species. In PE membranes, the cation species in L3 showed the presence of β_ca_ > 0, attributed to the cation–membrane interaction, in the range of high anisotropy of the membrane and the consequent highly anisotropic cross-sectional shape of paths (Figure 12). The fact that the β_ca_ values can be estimated indicates that the echo decays of the solvated cations in L3 almost follow random walk diffusion. On the other hand, the solvation structures of Li(EC)_3_(DEC)^+^ and Li(EC)_2_(DEC)_2_^+^ of L3 would have higher surface charge density compared with Li(EC)_4_^+^ of L3 due to the presence of weakly coordinated non-polar DEC species. That is the cause of larger β_ca_ of L3 than that of L1, as shown in Figure 12d. This result shows that the positive charge of Li cations, which is the cause of β_ca_, does not necessarily contribute to the restricted diffusion behavior of the lithium cations. Therefore, the specific restricted diffusivity of lithium cations is primarily associated with its solvation structure rather than the positive charge of the cations.

## 4. Design Concept of Ionic Conductors under the Influence of Solid Media

What should be emphasized through the series of research so far is that the ionic mobility responsible for the power performance of batteries is not determined by the electrolyte alone but is significantly affected by the solid media such as a separator and electrodes in the battery system. There are several types of interactions between the ions and solid media associated with the structure/morphology and chemical properties of carrier ions and solid media. Therefore, battery design is the process of optimizing the structure and performance of each battery element material and selecting the best combination of the materials to assemble a battery system with the expected performance using the indices of ionic mobility and its elemental factors.

From a microscopic point of view, the ionic mobility in the battery is determined by the magnitude of interactions between (1) the target ions and the species or sites around the ions in the electrolyte, and (2) the target ions and solid media such as the separator and electrodes surrounding the electrolyte in the battery. The interactions between (1) and (2) are not necessarily independent and may be correlated. As a result, there would be a case where the order of magnitude of ionic mobility of two types of electrolytes becomes opposite to that even in the same separator due to a specific interaction between the ion and either separator.

Due to the restriction as the battery structure is defined for proceeding the efficient and safe electrochemical reactions, it is unavoidable for carrier ions to be influenced by the solid media. Among solid media, the separator is an element material that has a great influence on the ionic mobility in the battery. This is because most of the electrolyte solution is fundamentally held in the separator, and the ions in it reciprocate between the cathode and the anode according to the charge-discharge reaction. From the viewpoint of an ion transport field in the battery, it is ideal for the separator to have low resistance in itself and not to interfere with ion movement, or rather to have the ability to enhance the ionic mobility. In addition, the porous structure of the separator membranes, in which the ions migrate in the linked pore spaces, could be controlled systematically by the structural factors as pore size, porosity, and path tortuosity that are significantly correlated with the ionic mobility in the pore-linked paths.

Individual evaluation of the microviscosities attributed to their interactions suggests the specific direction of size and properties of the paths for efficient ion movement in liquid/solid media. It can be said simply that reducing the interaction between the carrier ions and the separator membrane, which leads to reducing the microviscosities, is one of the important issues in advancing the battery design.

It is known well that the separator is also responsible for safety performance, such as a shut-down function. Its role is separate from ion mobility and, although not mentioned here, is important as another evaluation axis of separator membranes.

## 5. Conclusions

Ionic mobility in separator membranes and polymer gel electrolytes were evaluated using a new analytical approach developed, and the structural factors of the membranes and gel network of gel electrolytes responsible for ionic mobility quantitatively investigated through the estimation of microviscosites. In a battery device in which multiple elemental materials coexist, the obtained results that the determining factors of ion mobility can be elucidated in relation to the structural factors of the solid media is valuable, leading to the design of new materials and device of battery systems.

This study, which emphasized the importance of evaluating the interactions between the carrier ions and solid media in batteries, must continue to the evaluation of ion intercalation into the electrodes and ion migration in the electrode material to reach the charge transfer reaction sites, as a next step. After that, it could be clarified which location of ion migration in the battery is dominant for the high-rate performance of the battery.

Furthermore, secondary batteries inevitably deteriorate in performance with time with the progress of charge/discharge cycles. Since the types of constituent materials and operating conditions of batteries differ depending on their intended use, there are various degradation states of batteries as well as their responsible factors. Unfortunately, we have not so far evaluated the change in ion dynamics of batteries during the degradation process. In order to maximize the performance of each battery under each used condition, it is important not only to see the initial performance before degradation but also to follow the changes in mobility and microviscosity with the progress of charge-discharge cycles of the battery to design the battery system including the time axis.

Lithium secondary batteries and fuel cells are now indispensable devices for solving the problems of energy shortage and environmental pollution [34,55]. In order to accelerate the developments of the active devices, ion dynamics responsible for the high-power performance as well as the safety performance of those devices should be evaluated systematically based on the rational approach introduced here and others [56].

## Figures and Tables

**Figure 1 membranes-11-00277-f001:**
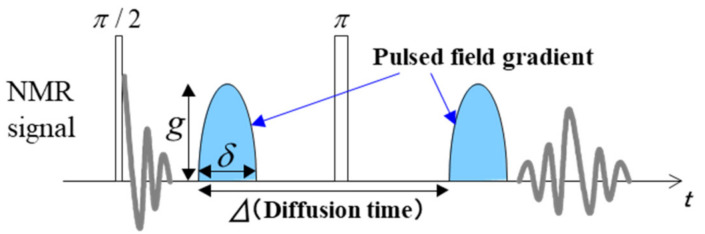
Typical pulse sequence for the diffusion coefficient measurement. In the evaluation of the T_2_ behavior, an echo signal with δ = 0 (without the application of the gradient pulse).

**Figure 2 membranes-11-00277-f002:**
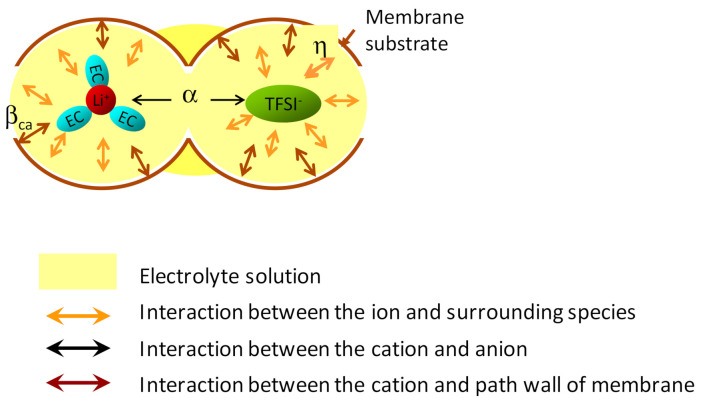
Schematic diagram of interactions on the ions of the solution located in the space of the separator membrane. η; microviscosity attributed to the van der Waals interaction between the ions (Li(EC)_3_^+^, TFSI^−^) and all surrounding species in the solution (yellow region), α; microviscosity from the Coulombic interaction between the closest cations and anions (which is a dominant element from the Coulombic interactions), β_ca_; microviscosity reflecting the Coulombic interaction between the cations and inside wall of pores (brown barrier) that holds the solution. Reprinted with permission from [24]. Copyright 2017 ACS Publications.

**Figure 3 membranes-11-00277-f003:**
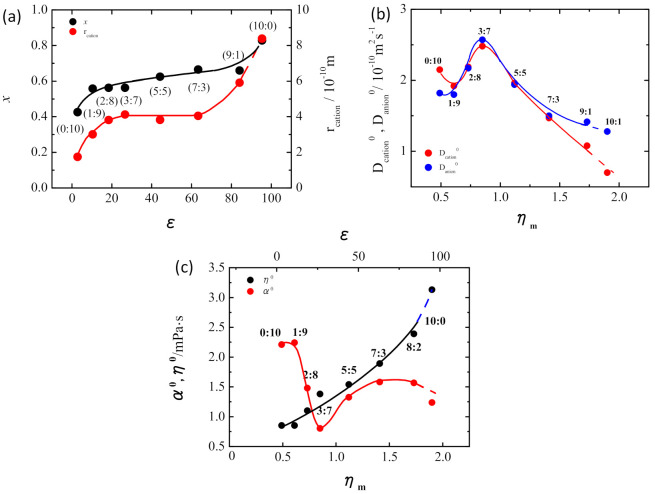
(**a**) Dissociation degree of the salt, *x* and the radius of the cation species, r_cation_, (**b**) inherent diffusion coefficients of the cation, *D*^0^_cation_, and anion, *D*^0^_anion_, and (**c**) microviscosities, η^0^ and α^0^, of the 1 M electrolyte solution, lithium bis(trifluoromethanesulfonyl)imide/ethynatelene carbonate + dimethyl carbonate (LiTFSI/EC+DMC) with changing the wt. mixing ratio of (EC:DMC) as shown in the figure, as functions of macroscopic dielectric constant, *ε* and bulk viscosity, η_m_ of the solution. The lithium salt, LiTFSI, was selected in this research due to the higher stability in the solvent to obtain reliable comparison data. Reprinted with permission from [17]. Copyright 2012 ACS Publications.

**Figure 4 membranes-11-00277-f004:**
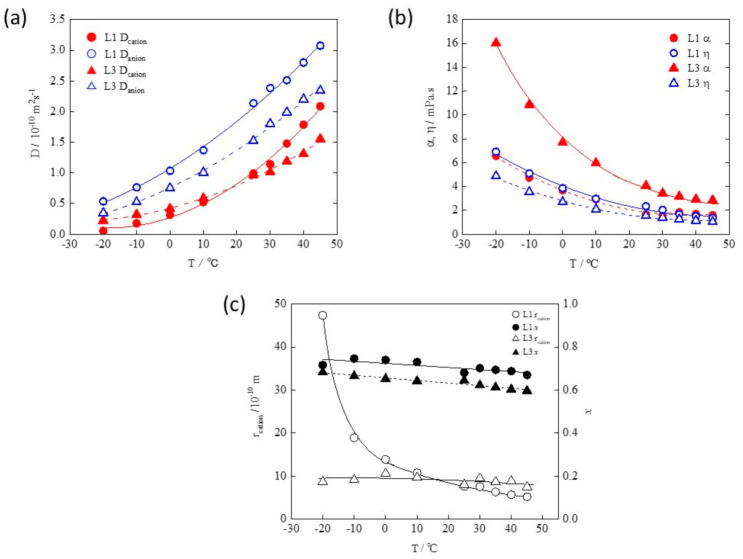
Temperature dependences of (**a**) *D*_cation_ and *D*_anion_, (**b**) η and α, and (**c**) *r*_cation_ and *x* of L1; 1 M LiPF_6_ /EC+DEC (EC:DEC = 1:1) and L3; 1 M LiPF_6_ /EC+DEC (EC:DEC = 1:6.9) solutions without the separator membrane. Reprinted with permission from [24]. Copyright 2017 ACS Publications.

**Figure 5 membranes-11-00277-f005:**
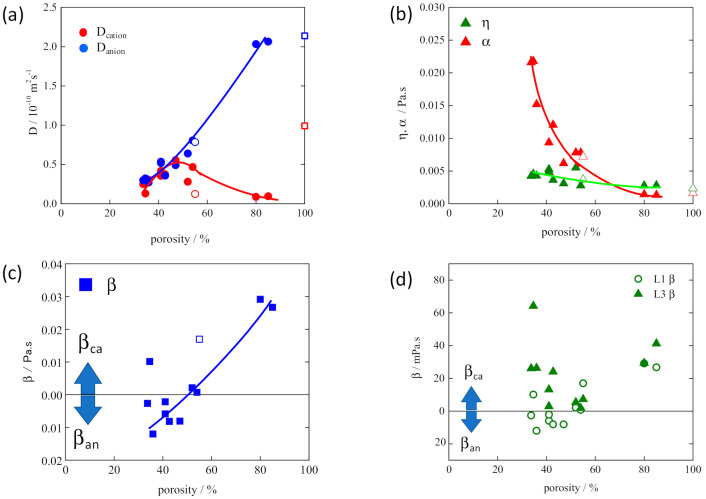
Estimated dynamic values: (**a**) *D*_cation_, *D*_anion_ (**b**) η, α (**c**) β_cation_, β_anion_ of L1 solution in separator membranes, and (**d**) comparison of β_cation_, β_anion_ between L1 and L3 as a function of porosity of the membrane. Hallow marks at 55% represent the polypropylene (PP) membrane and at 100% represent the free electrolyte solution. Reprinted with permission from [24,30]. Copyright 2017, 2018 ACS Publications.

**Figure 6 membranes-11-00277-f006:**
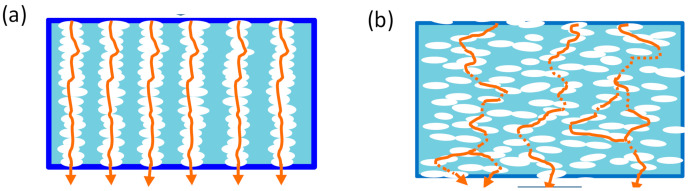
Schematic diagram of the cross-section of separator membrane with (**a**) linear pathways across the membrane surface by the alignment of pores having the tortuosity ~1, and (**b**) randomly arranged pathways with the tortuosity > >1. Reprinted with permission from [33]. Copyright 2018 ACS Publications.

**Figure 7 membranes-11-00277-f007:**
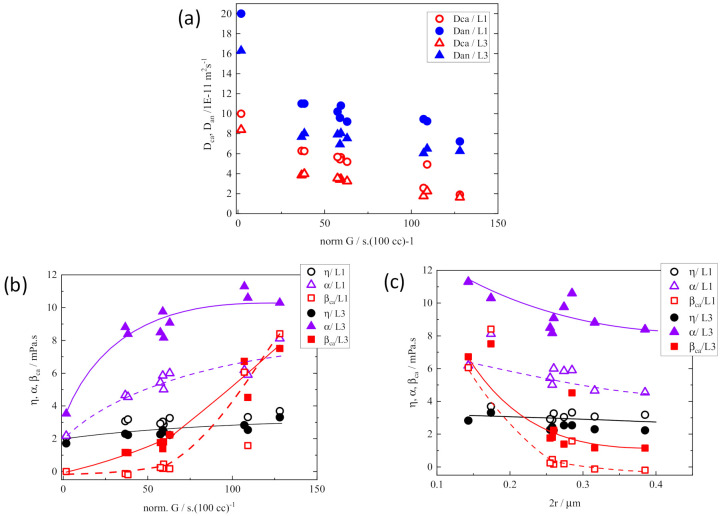
(**a**) *D*_ca_ and *D*_an_ with respect to normalized *G* (=*G*′), and microviscosities, η, α, β_ca_ of the ionic species with respect to (**b**) normalized *G* (=*G*′) and (**c**) average pore diameter of L1 and L3 solutions in the PP separator membranes. Reprinted with permission from [33]. Copyright 2018 ACS Publications.

**Figure 8 membranes-11-00277-f008:**
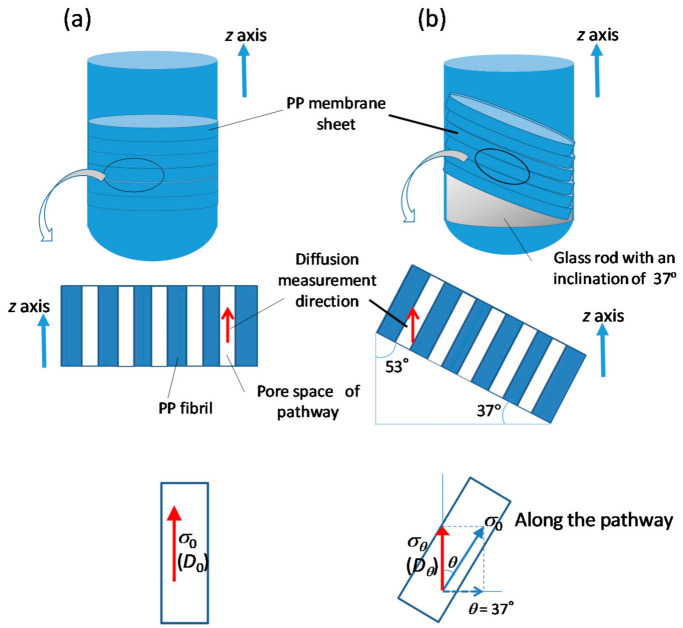
Schematic illustrations showing the direction of diffusion with respect to the straight pathway of the aligned-pore membranes: (**a**) vertically stacked polypropylene (PP) membrane sheets; (**b**) PP membranes inclined at an angle of 53° from the longitudinal axis (z-axis) of the sample tube. The z-axis corresponds to the diffusion measurement direction. The middle illustrations represent the internal structure of the aligned-pore membrane with straight pathways: (blue) PP stem fibril; (white) pore space for ion movement. The lower illustration represents the diffusion components that are detected (red arrows) and the respective conductivity components. In the case of (b), *σ**_θ_* = *σ*_0_ cos^2^
*θ* is satisfied when *D**_θ_* = *D*_0_ cos^2^
*θ*. If the effect of ionic collision with the wall on the diffusivity in the inclined direction is larger than that on *D*_0_, *D**_θ_* < *D*_0_ cos^2^
*θ* and the subsequent relationship *σ_θ_* < *σ*_0_ cos^2^
*θ* are satisfied. Reprinted with permission from [37]. Copyright 2019 ACS Publications.

**Figure 9 membranes-11-00277-f009:**
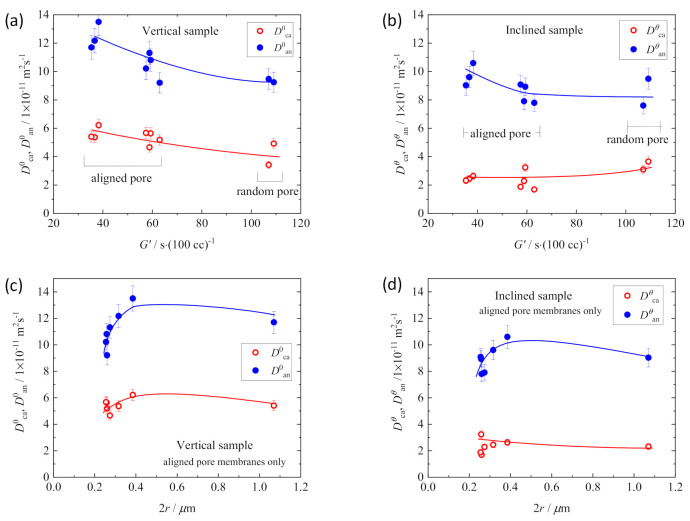
Estimated *D*_ca_ and *D*_an_ of L1 solution in (**a**,**c**) vertical PP membranes (identified by superscript 0) and (**b**,**d**) inclined PP membranes (identified by superscript *θ*) as functions of (**a**,**b**) normalized *G* (*G*′) and (**c**,**d**) pore diameter (= path width) (2*r*). Reprinted with permission from [37]. Copyright 2019 ACS Publications.

**Figure 10 membranes-11-00277-f010:**
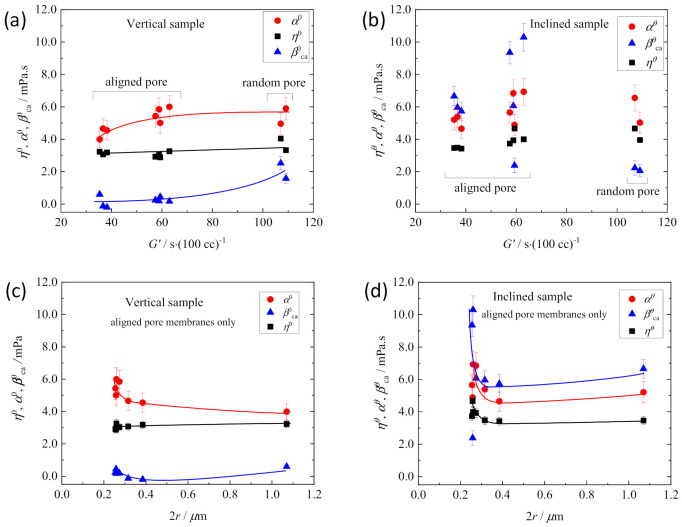
Estimated microviscosities, η, α, and β_ca_ of L1 solution in (**a**,**c**) vertical PP membranes and (**b**,**d**) inclined PP membranes as functions of (**a**,**b**) *G′* and (**c**,**d**) 2*r*. Reprinted with permission from [37]. Copyright 2019 ACS Publications.

**Figure 11 membranes-11-00277-f011:**
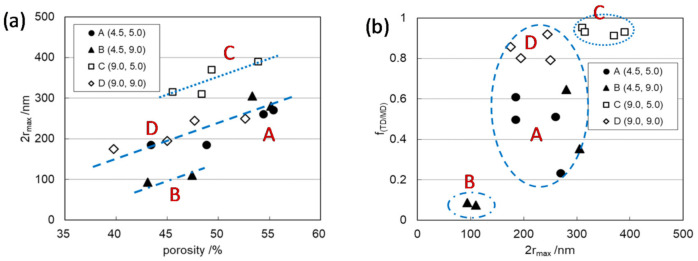
(**a**) Change in mode pore size (2*r*_max_) as a function of polyethylene (PE) membrane porosity, and (**b**) correlation between the anisotropy index (*f*_(TD/MD)_) evaluated by the elastic modulus in machine direction (MD) and transverse direction (TD) and the mode pore size, 2*r*_max_. The display in the figure represents the four groups of membranes (A, B, C, D) prepared in the process of different stretching conditions: e.g., A (4.5, 5.0); membranes prepared with the stretching ratios of MD = 4.5 and TD = 5.0. Reprinted with permission from [15]. Copyright 2020 ACS Publications.

**Figure 12 membranes-11-00277-f012:**
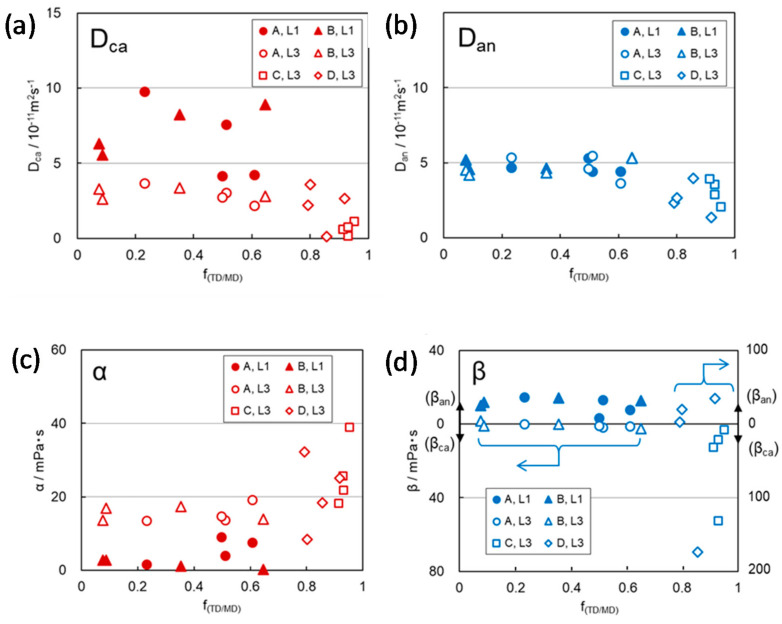
Estimated (**a**) *D*_ca_, (**b**) *D*_an_, (**c**) α, and (**d**) β_ca_, β_an_ of PE membranes of A and B groups using L1 and L3 solution and the membranes of C and D groups using L3 solution as a function of anisotropy index, *f*_(TD/MD)_. Reprinted with permission from [15]. Copyright 2020 ACS Publications.

**Figure 13 membranes-11-00277-f013:**
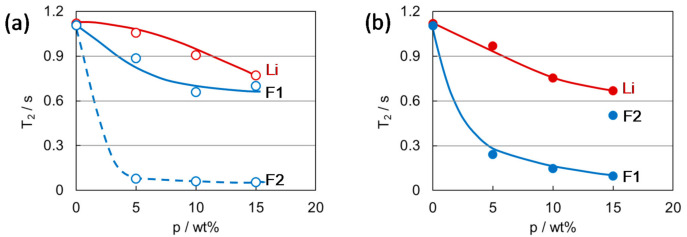
*T*_2_ (Li) (red symbol) and *T*_2_ (F) (blue symbol) change with the polyvinylidene difluoride (PVDF) polymer fraction of the gel for (**a**) annealed and (**b**) quenched gels. *T*_2_ (F) of the annealed and quenched gels at 15 wt% of PVDF were estimated by two-component fitting: *T*_2_ (F1) is the major component (>80%), while *T*_2_ (F2) is the minor component (<20%). The electrolyte solution used for gelation is 1 M LiTFSI/PC, in which LiTFSI was selected to reduce the deterioration of the electrolyte by gelation and mono-solvent. PC was selected to simplify the gel structure. Reprinted with permission from [42]. Copyright 2019 ACS Publications.

**Figure 14 membranes-11-00277-f014:**
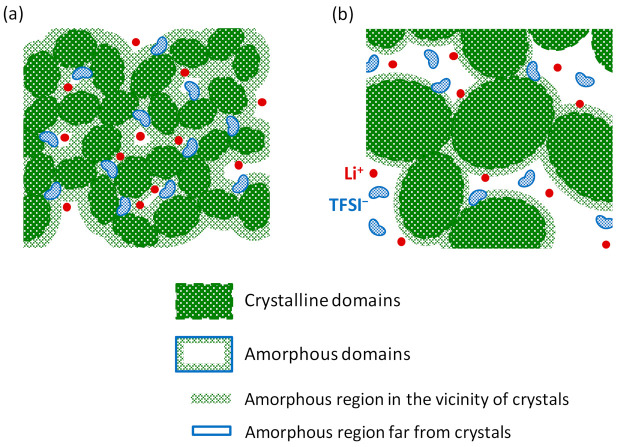
Schematic features of PVDF gels composed of crystalline and amorphous phases with (**a**) smaller crystals formed via the quenching process or via the annealing process and (**b**) larger crystals formed via the annealing process. Reprinted with permission from [14]. Copyright 2020 ACS Publications.

**Figure 15 membranes-11-00277-f015:**
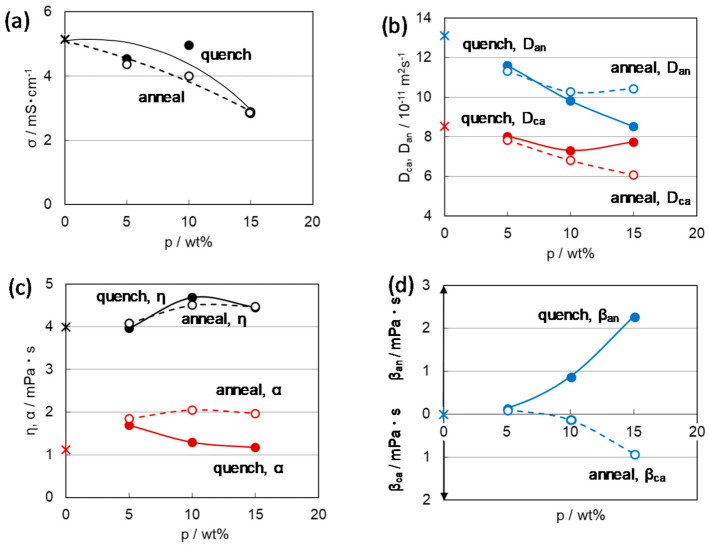
(**a**) Ionic conductivities, (**b**) *D*_ca_, *D*_an_ (**c**) η, α (**d**) β_ca_, β_an_ of the gel composed of 1 M LiTFSI/PC and PVDF as a function of the PVDF polymer fraction in the gel. The values at *p* = 0 are for the electrolyte solution without PVDF. Reprinted with permission from [42]. Copyright 2019 ACS Publications.

**Figure 16 membranes-11-00277-f016:**
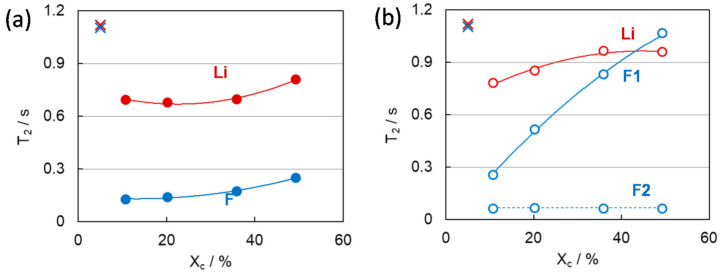
*T*_2_ (Li) (red) and *T*_2_ (F) (Blue) of the gels with 10 wt% of PVDF-based polymer and 90 wt% 1 M LiTFSI/PC prepared by (**a**) quenching and (**b**) annealing. F1 is the value of the major component (>80%), and F2 is the values of the minor component (<20%). × symbols represent the *T*_2_ (Li) (red) and *T*_2_ (F) (blue) of the free electrolyte solution without polymer. Reprinted with permission from [14]. Copyright 2020 ACS Publications.

**Figure 17 membranes-11-00277-f017:**
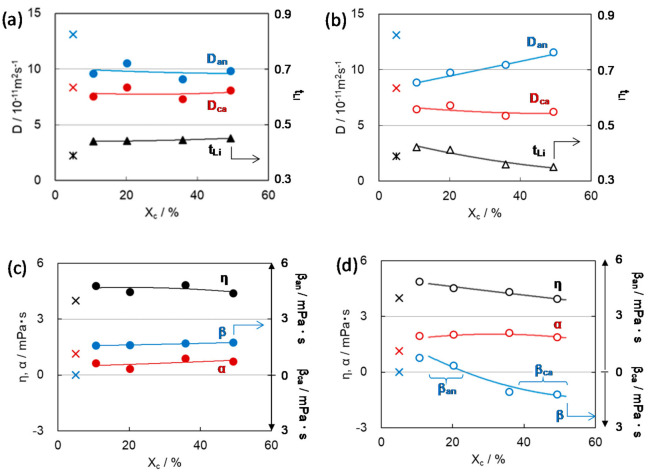
*D*_ca_, *D*_an_, cation transport number (*t*_Li_), and η, α, β of the gels composed of 90 wt% of 1 M LiTFSI/PC and 10 wt% of PVDF-based polymer prepared by (**a**,**c**) quenching and (**b**,**d**) annealing. The values of the free electrolyte solution without polymer are shown by × symbols. Reprinted with permission from [14]. Copyright 2020 ACS Publications.

**Figure 18 membranes-11-00277-f018:**
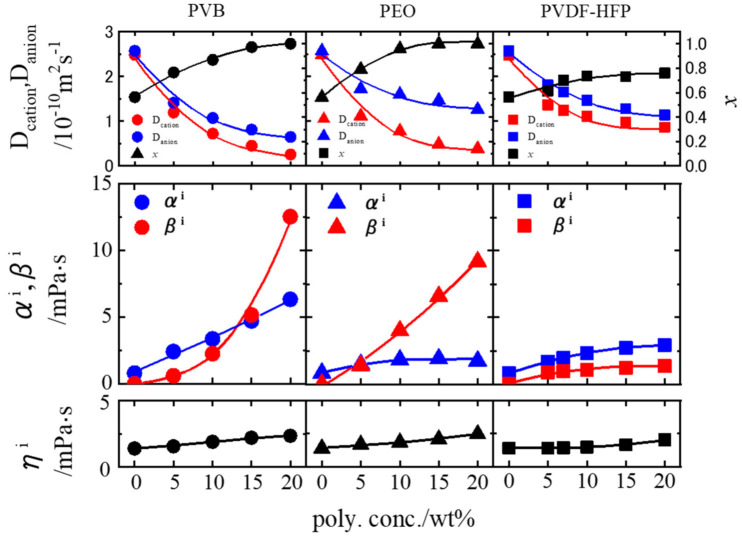
D_ca_, D_an_, α, β, and η of poly(vinyl butyral) (PVB), poly(ethylene oxide) (PEO), and poly(vinylidene fluoride-hexafluoropropylene) (PVDF-HFP) polymer gel electrolytes with changing the polymer fraction in the gels. 1 M LiTFSI/(EC+DMC(3:7)) and each polymer of each fraction were mixed at 60–70 ℃ to complete the homogeneous solution and annealed for gelation. Reprinted with permission from [17]. Copyright 2012 ACS Publications.

**Figure 19 membranes-11-00277-f019:**
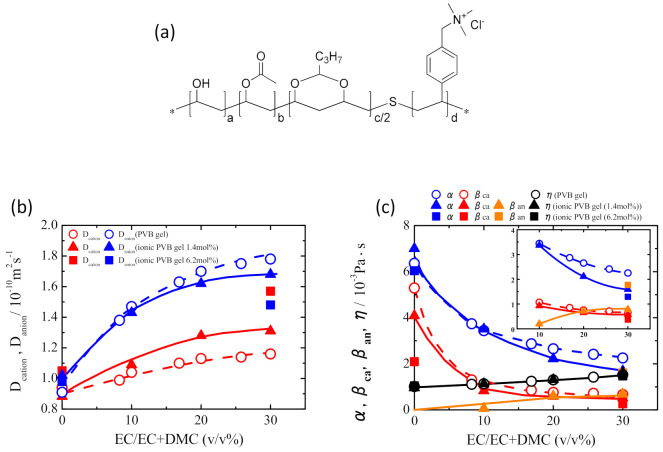
(**a**) Chemical structure of the ionic PVB polymer. (**b**) *D*_ca_ (red symbols) and *D*_an_ (blue symbols) and (**c**) η, α, β_ca_, β_an_ of the PVB gel (○) and ionic PVB gels (▲; 1.4 mol%, ■; 6.2 mol%) with 1 M LiTFSI/EC + DMC as a function of EC fraction of the binary solvent. EC fraction change is associated with the change in the solvation structure of lithium cations. Reprinted with permission from [18]. Copyright 2014 ACS Publications.

**Figure 20 membranes-11-00277-f020:**
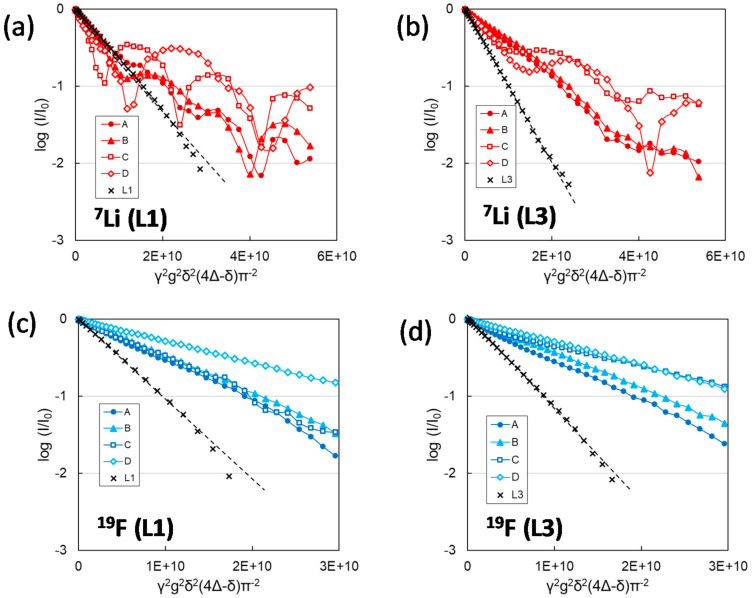
Comparison of the ^7^Li (**a**,**b**) and ^19^F (**c**,**d**) echo attenuations of L1 (**a**,**c**) and L3 (**b**,**d**) solutions in PE membranes at different stretching ratios of MD to TD used to prepare the porous membranes. A = 4.5:5.0, B = 4.5:9.0, C = 9.0:5.0. The black symbols (×) and dashed lines represent the echo decay of free electrolyte solutions, L1 and L3. Reprinted with permission from [51]. Copyright 2020 ACS Publications.

**Table 1 membranes-11-00277-t001:** Determining factors of ionic mobility in electrolytes in separator membranes and polymer gel electrolytes.

	Carrier	Solid Media	
	Li^+^ Cation	Separator	Gel Electrolyte
Solvation structure	Exchange reaction of solvent ligands		
Physical structure		Porosity Pore size Path tortuosity Cross-sectional shape of path	Polymer crystalline domain size (in PVDF gel)
Chemical structure		Polar groups Adsorbed species Surface charge on path wall	Polar groups on polymer chains

## Data Availability

This research follows MDPI Research Data Policies.

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
