# Peer review of "Ion Transport in Solid Medium—Evaluation of Ionic Mobility for Design of Ion Transport Pathways in Separator and Gel Electrolyte"

_membranes, 2021, doi:10.3390/membranes11040277_

Round 1
Reviewer 1 Report
The separator membrane in a battery system is an indispensable element to keep separation between the cathode and anode for prevention of their short circuit, and to hold the electrolyte solution inside safely with delivering ions to electrodes for charge transfer reactions of energy supply. Besides, Further improvement in the performance of lithium secondary batteries will be an indispensable issue to realize a decarbonized society. Among them, the batteries for electric vehicles have still many issues to be addressed because they are subject to various conditions such as high power performance, safety, and cost restrictions for widespread use. Those subjects require extensive researches from improvement of each element material to control the battery system to optimize the total performance. In this paper, the author introduced the evaluation approach of ion dynamics, and the evaluation results mobility and interactive situations of carrier ions in the practical separator membranes and gel electrolytes. Finally, the direction of material design was outlined through this review. I am pleased to send you moderate comments. The results and theme of this paper is quite interesting. The layout is clear and easy to understand. Generally, this manuscript makes fair impression and my recommendation is that it merits publication in this Journal, after the following major revision:
- The detailed literature review indicates efforts made by the authors. The coherence of the related work, however, is still not clear. It may help the authors by answering the following questions: Why are these works relevant? Which specific problems were addressed? How are the previous results related with the latest work? What are the outstanding, unresolved, research issues? Which of them has been solved by the proposed study? Answering the questions leads to the novelty of the proposed work naturally. Besides, the current one is nothing but a literature review. Why their work is important comparing to previous reports? I think this is essential to keep the interest of the reader.
- Please check all Equations double.
- In Fig.3, 4, 5 and 7, 9-13, 15-20, the authors should give the explanations for the difference of data collected from different sources.
- There is no CONCLUSION part for the manuscript.
- Energy shortage and environment pollution have seriously threatened people’s survival. Thus, the development of battery and fuel cell has caught human attention. Besides lithium secondary batteries, fuel cells have attracted attention from energy devices such as portable, mobile and stationary devices, since it helps effective reductions of energy shortage and environment pollution, see [International Journal of Hydrogen Energy, 2018, 43(37):17880-17888; Fractals, 2019, 27(2):1950012; J. Power Sources 2007, 164, 351-364; Membranes 2021, 11(3), 168; Membranes 2021, 11(3), 159].
- Please, expand the conclusions in relation to the specific goals and the future work.
Reviewer 2 Report
Saito et al. has elucidated the mechanism of ionic mobility from the microscopic point of view giving its identification and correlation between viscosity and diffusion coefficients of electrolytes. Also, the effect of solid media where ionic movement takes place is discussed with evaluation of ion dynamics and interactive situations of ions. It is valuable that this article suggested a guidance for setting up the practical environment of electrolyte system. However, there are some typos and mistakes needed to be modified in the manuscript. Therefore, in my opinion, to be accepted for the publication in Membranes journal, a minor revision process is required. Here are the minor points.
- There are too many self-citations in the whole contents. In my humble opinion, there might be more researches other groups have conducted, which can support the author’s work. Please consider possible citations.
- Page 6 /line 239 and Page 10 / lien 330
The numbering order of subsections is repeated. Please check if the second subsection is correct. The lettering of each subsection is also required to be rearranged depending on the numbering order of subsections.
- Page 10 / line 347
The caption of table should be located below the table. Moreover, its format seems to need reorganization by using bullet points.
- Page 11 / line 377
Please check if Dantion is correct. It looks like a typo.
- Page 12 / line 398-399
This sentence is divided into two lines. Please correct this error.
Without above minor points, I believe the author’s work is worthwhile to be published in Membranes journal since this article offers much information of chemical and physical properties in each condition.
Reviewer 3 Report
Dear author,
My sincere thanks for the elaborated paper with many details and aspects discussed on separator technology for Li-ion.
Some minor issues only I like to discuss:
Line 106: 100mV is rather high; normally maximal 10mV is used as perturbation for the AC voltage
For the whole review: Please include a list of parameters including their dimension
Line 323: Where is the proof for the second and third hydration shell?
Line 366: inhibits
Line 374: site
Line 398-399: delete empty space between the ......low
Line 429: motility? Use "mobility"
Line 459: poruous? Use "porous"
Line 499: lager? Use "larger"
Line 587: motility? Use "mobility"
Line 627: caition? Use "cation"
Line 657: cross-linked
Then a general remark:
> Li-ion batteries and Li-metal batteries may use a shut-down separator with a 3-layer structure. I see no discussion about this. Could you discuss shortly?
> Li-ion batteries: the C-rate depends on the Solid Electrolyte Interface resistance and the intercalation rate of the anode. This was not discussed. Please discuss this very shortly at the end of the paper as nowadays hybrid Li-ion batteries (activated carbon anode and intercalation type of cathode) have very high specific power at high specific energy values. The separator seems not be to the limiting part then but more the anode intercalation and the SEI layer. Please describe this shortly and if possible dedicate a new study towards this (another paper).
The given corrections are minor and easily corrected.
Please give at the end of the paper also a short overview of commercially available Li-in and Li-metal type separators and their area resistance (in Ohm.cm2) to indicate their impact for a 1 M Li-salt like LiPF6.
Kind regards,
Reviewer
Round 2
Reviewer 1 Report
The revised form is still not acceptable for publication, since the authors don't follow none of my comments and even make it worse.
Author Response
thanks